# Low HDL Cholesterol Efflux Capacity Indicates a Fatal Course of COVID-19

**DOI:** 10.3390/antiox11101858

**Published:** 2022-09-21

**Authors:** Julia T. Stadler, Harald Mangge, Alankrita Rani, Pero Curcic, Markus Herrmann, Florian Prüller, Gunther Marsche

**Affiliations:** 1Division of Pharmacology, Otto Loewi Research Center for Vascular Biology, Medical University of Graz, Universitätsplatz 4, 8010 Graz, Austria; 2Clinical Institute of Medical and Chemical Laboratory Diagnostics, Medical University of Graz, Auenbruggerplatz 15, 8036 Graz, Austria

**Keywords:** COVID-19, HDL, cholesterol efflux capacity, LCAT, anti-oxidative capacity, PON1

## Abstract

Plasma membrane cholesterol is required for proper trafficking and localization of receptors that facilitate severe acute respiratory syndrome coronavirus 2 infection. High-density lipoproteins (HDL) mobilize plasma membrane cholesterol, and HDL-cholesterol levels are associated with the severity of COVID-19 disease and mortality. However, HDL-cholesterol levels poorly reflect the function of this complex family of particles, and a detailed assessment of COVID-19-associated changes in HDL functionality and its prognostic value is lacking. In the present study, we assessed HDL cholesterol efflux capacity, HDL anti-inflammatory and antioxidant properties, and changes in HDL composition and metabolism in COVID-19 (*n* = 48) and non-COVID pneumonia patients (*n* = 32). COVID-19 infection markedly reduced the activity of lecithin-cholesteryl-acyltransferase and functional parameters of HDL, such as the cholesterol efflux capacity, arylesterase activity of paraoxonase 1, and anti-oxidative capacity of apoB-depleted serum when compared to non-COVID pneumonia at baseline, paralleled by markedly reduced levels of HDL-cholesterol. Of particular interest, low HDL cholesterol efflux capacity was associated with increased mortality risk in COVID-19 patients, independent of HDL-C levels. Our results highlight profound effects of COVID-19 infection on HDL function, metabolism, and composition. Low HDL cholesterol efflux capacity indicates a fatal course of COVID-19, independent of HDL-cholesterol levels.

## 1. Introduction

The severe acute respiratory syndrome coronavirus 2 (SARS-CoV-2) is the causative agent of the severe respiratory illness known as 2019 coronavirus disease (COVID-19). This disease primarily infects the respiratory tract, but a multitude of organs throughout the body are also affected [1].

It is well known that elevated serum levels of low-density lipoprotein cholesterol (LDL-C) and low levels of high-density lipoprotein cholesterol (HDL-C) increase the risk of cardiovascular disease [2,3]. On the contrary, recent evidence suggests that low serum levels of LDL-C, HDL-C, and total cholesterol at hospital admission are associated with severe disease and mortality in COVID-19 patients. Specifically, a reduction of serum HDL-C in COVID-19 patients is linked with disease severity [4,5,6,7,8,9,10]. Moreover, patients with higher antecedent HDL-C levels have a lower risk of COVID-19 infection [11,12,13].

Preclinical evidence suggests that HDL particles play an important role in regulating inflammation, with potential mechanisms including impeding viral fusion, neutralizing exaggerated immune responses, and reducing the rate of bacterial complications [14,15]. The cholesterol efflux capacity of HDL is directly linked to the ability of HDL to modulate plasma membrane cholesterol content [14,16]. This is of particular importance, given the recent evidence that plasma membrane cholesterol increases the trafficking of angiotensin-converting enzyme 2 (ACE2) to lipid rafts, thereby increasing SARS-CoV-2 infection [17,18].

HDL particles exhibit great heterogeneity in structure, size, and composition, which affects their role in regulating inflammation and neutralizing exaggerated immune responses [15,19,20]. During acute or chronic inflammation, profound alterations in the composition and functionality of HDL particles have been reported [21,22,23,24,25]. HDL-C levels do not reflect the function of these complex particles, and HDL functions correlate only weakly with serum HDL-C concentrations. Therefore, direct measurements of HDL function are needed rather than relying on surrogate markers such as the concentration of HDL-C.

The effect of COVID-19 on HDL function remains poorly characterized, and only a few studies have investigated the composition and functionality of HDL in COVID-19 patients. One study reported alterations in the HDL proteome of COVID-19 patients [26], and another study observed a lower endothelium-protective capacity of HDL during COVID-19 infection when compared to healthy subjects [27].

In the present study, we assessed multiple metrics of HDL function in COVID-19 patients compared to patients having non-COVID-19 pneumonia. We further determined the potential of HDL-related parameters as prognostic markers for COVID-19-associated severity and risk of death.

## 2. Materials and Methods

### 2.1. Study Population and Study Design

We established a biobank (Alpe_Adria_Coronavirus_Cohort, ALDOCOV) by collecting leftovers of blood samples from patients suffering from COVID-19 whenever they were sent to the central laboratory of our University hospital during the period April–December 2020. After the completion of all routine laboratory testing, the residual material was stored at −80 °C until batched analysis. In this retrospective study, we measured the plasma concentrations of interleukin-6 (IL-6), C-reactive protein (CRP), and creatinine. Basic clinical characteristics and anteceding diseases, including cardiovascular, oncologic, renal, hypertension, pulmonary, and metabolic (diabetes, obesity), were recorded in the database. Anthropometric and clinical data, as well as outcome data, were obtained from the laboratory and hospital information systems. The primary outcome was death within 90 days after admission. Respiratory support with oxygen was used as the secondary endpoint. Of the 48 COVID-19 patients, the primary endpoint, death, occurred in 31% of the study cohort. The institutional ethics committee of the Medical University of Graz (EK 32-475 ex 19/20) approved this study.

### 2.2. Laboratory Measurements

Plasma lipids such as total cholesterol, triglycerides, and HDL-C were measured by enzymatic photometric transmission measurement (Roche Diagnostics, Mannheim, Germany). The concentrations of LDL-cholesterol were calculated by the Friedewald’s formula [28]. Interleukin-6 (IL-6) and C-reactive protein (CRP) were measured with commercial immunoassays on a COBAS 8000 analyzer (Roche Diagnostics, Rotkreuz, Switzerland).

### 2.3. Preparation of apoB-Depleted Serum

ApoB-depleted serum was prepared by adding 40 μL polyethylene glycol (20% in 200 mmol/L Glycine buffer) (Sigma-Aldrich, Darmstadt, Germany) to 100 μL serum followed by gentle mixing. The serum was incubated at room temperature for 20 min and centrifuged at 10,000× *g* for 20 min at 4 °C. Subsequently, the supernatant was collected, and the samples were stored at −70 °C until use.

### 2.4. Cholesterol Efflux Capacity

Cholesterol efflux capacity of apoB-depleted serum was assessed, as described [29,30]. Briefly, J774.2 cells (Sigma-Aldrich, Darmstadt, Germany) were maintained in DMEM medium (Life Technologies, Carlsbad, CA, USA), then supplemented with 10% FBS and 1% penicillin/streptomycin (P/S). Cells were seeded on 48-well plates (Greiner Bio-One, Kremsmünster, Austria) (300,000 cells/well), cultured for 24 h, and labeled with 0.5 μCi/mL radiolabeled [^3^H]-cholesterol (Hartmann Analytic, Braunschweig, Germany) in DMEM supplemented with 2% FBS and 1% P/S in the presence of 0.3 mM 8-(4-chlorophenylthio)-cyclic AMP (Sigma-Aldrich, Darmstadt, Germany) overnight. After labeling, cells were rinsed with serum-free DMEM containing 1% P/S and equilibrated with serum-free DMEM containing 1% P/S and 2 mg/mL bovine serum albumin (Sigma-Aldrich, Darmstadt, Germany) for 2 h. Subsequently [^3^H]-cholesterol efflux was determined by incubating cells for 3 h with 2.8% apoB-depleted serum. Cholesterol efflux was expressed as radioactivity in the cell culture supernatant relative to total radioactivity of the cell culture supernatant and cells. All steps were performed in the presence of 2 μg/mL of the acyl coenzyme A cholesterol acyltransferase inhibitor Sandoz 58-035 (Sigma-Aldrich, Darmstadt, Germany).

### 2.5. LCAT Activity

LCAT activity of serum was measured in duplicates, using a commercially available kit (Merck, Darmstadt, Germany) according to the manufacturer’s instructions. Specifically, samples were incubated with LCAT substrate for 4 h at 37 °C. The fluorescent substrate emits fluorescence at 470 nm. When the substrate is hydrolyzed by LCAT, a monomer is released that emits fluorescence at 390 nm. The LCAT activity is assessed over time and is expressed in changes of 470/390-nm emission intensity [30].

### 2.6. Arylesterase Activity of Paraoxonase

Ca^2+^-dependent arylesterase activity of PON1 was determined by a photometric assay using phenylacetate substrate, as described [25]. Activities were calculated from the slopes of the kinetic chart of two independent experiments, measured in duplicates.

### 2.7. Anti-Oxidative Capacity

The anti-oxidative activity of apoB-depleted serum and isolated HDL was determined as previously described [31]. Briefly, dihydrorhodamine (DHR) was suspended in DMSO to a 50 mM stock, which was diluted in HEPES (20 mM HEPES, 150 mM NaCl, pH 7.4) containing 1 mM 2,2′-azobis-2-methyl-propanimidamide-dihydrochloride (Sigma-Aldrich, Darmstadt, Germany) to a 10 μM working reagent. 10 µL of apoB-depleted serum (1:10 diluted) was placed in 384-well plates (Greiner Bio-One, Kremsmünster, Austria) and the volume was adjusted to 100 μL with HEPES buffer containing 10 μM DHR. The increase in fluorescence due to oxidation of DHR was monitored for 90 min at 538 nm. The increase in fluorescence per minute of DHR in the absence of apoB-depleted serum or isolated HDL was set to 100%, and individual apoB-depleted serum samples were calculated as percentage of inhibition of DHR oxidation.

### 2.8. Anti-Inflammatory Activity by Inhibition of NFkB Expression

U937 monocytic cells containing a 5× NF-κB green fluorescence protein (GFP) reporter cassette were cultivated in RPMI 1640 (Life Technologies, Carlsbad, CA, USA) containing 10% fetal bovine serum (FBS) (Life Technologies, Carlsbad, CA, USA) and 1% P/S. The cells were pre-incubated for 90 min with 7% apoB-depleted serum in duplicates. Subsequently, cells were stimulated for 24 h with lipopolysaccharide (LPS) (50 ng/mL) (Sigma, Darmstadt, Germany), collected by centrifugation at 400× *g* for 7 min, and fixed with 100 μL fixative solution, which was prepared as previously described [31]. The expression of NF-κB was assessed by flow cytometry.

### 2.9. Quantification of Serum Amyloid A (SAA)

Serum amyloid A (SAA) was quantified using a commercially available kit (Invitrogen, Carlsbad, CA, USA), according to the manufacturer’s instructions.

### 2.10. Statistical Analyses

All statistical analyses were performed with SPSS (Version 28.0.1.1) (SPSS, Inc., Chicago, IL, USA) and Graphpad Prism (Version 9.4.0). Continuous variables were summarized as medians (Q1–Q3) and absolute variables as absolute frequencies (%). Differences between groups were assessed using Mann–Whitney U test. Correlations were determined using Spearman’s correlation coefficient *rho* and were corrected according to Bonferroni. For survival analyses, Kaplan–Meier and Cox regression models were used to find associations between HDL-related parameters and mortality. The Cox regression model was adjusted for age, gender, and HDL-C levels.

## 3. Results

### 3.1. Baseline Characteristics of the Study Cohort

In this study, 48 COVID-19 patients were prospectively enrolled (Table 1). Non-COVID pneumonia patients, matched in age and gender, served as the control cohort. Serum levels of total cholesterol, LDL-C and HDL-C, were lower in COVID-19 patients, whereas triglyceride levels were similar in both cohorts. The inflammation marker C-reactive protein (CRP) and interleukin (IL)-6 did not differ between the two groups. In the COVID-19 cohort, 19% of patients did not receive any oxygen therapy, whereas 44% were treated by oxygen mask, 10% of patients with continuous positive airway pressure (CPAP), and 27% received mechanical ventilation. Forty percent percent of the 48 COVID-19 patients were treated in the intensive care unit, and 31% died within 90 days.

### 3.2. COVID-19 Is Associated with Alterations in HDL Metabolism and Parameters of HDL Function

We next assessed the effect of COVID-19 on HDL composition, metabolism, and function (Figure 1). HDL-C levels were significantly reduced in COVID-19 patients (Table 1, Figure 1A) when compared to non-COVID-19 pneumonia patients. We observed no significant difference in CRP (Table 1) or levels of HDL-associated acute phase protein serum amyloid A (SAA) (Figure 1G) between groups. SAA levels strongly correlated with CRP levels (r_S_ = 0.652, *p* < 0.001) (Figure 2).

To gain insights into the metabolism of HDL, the activity of the important enzyme lecithin-cholesteryl acyltransferase (LCAT) was assessed. LCAT converts free cholesterol into cholesteryl esters, making the newly synthesized HDL spherical [32,33]. We observed that LCAT activity was significantly decreased in the COVID-19 patients (*p* = 0.015) compared to non-COVID-19 pneumonia patients.

We next assessed functional parameters of HDL. The HDL cholesterol efflux capacity is a measure of the first step in reverse cholesterol transport, which is one of the best-studied functions of HDL [29,34]. HDL cholesterol efflux capacity was reduced in COVID-19 patients compared to the control group (*p* < 0.001), (Figure 1C). Moreover, the arylesterase (AE) activity of PON1 and the anti-oxidative (AO)-capacity of apoB-depleted serum were also lower in COVID-19 patients (*p* = 0.034, *p* < 0.001) when compared to non-COVID-19 pneumonia patients (Figure 1D,E). The ability of apoB-depleted serum to suppress monocyte NF-κB expression induced by lipopolysaccharide (anti-inflammatory activity) did not differ between groups (*p* = 0.245) (Figure 1F).

We further performed correlation analysis (Figure 2) of the assessed HDL-related parameters with clinical data from the COVID-19 cohort to provide an overview of the association with HDL-functionalities. Interestingly, patient age was negatively associated with AE-activity of PON1 after Bonferroni correction (r_S_ = −0.485, *p* > 0.001). Of particular interest, inflammatory markers, especially CRP, were negatively associated with HDL-C (r_S_ = −0.482, *p* = 0.001), HDL cholesterol efflux capacity (r_S_ = −0.585, *p* < 0.001), LCAT (r_S_ = −0.411, *p* = 0.007), and AE-activity of PON1 (r_S_ = −0.500, *p* = 0.001), demonstrating a significant impact of inflammation on HDL function. Moreover, we observed a positive correlation between creatinine and the AO-capacity of apoB-depleted serum (r_S_ = 0.432, *p* = 0.004).

### 3.3. HDL-Related Parameters and Treatment in COVID-19 Patients

Based on the treatment, COVID-19 patients were stratified into two groups: (1) no oxygen therapy, (2) oxygen therapy including treatment with oxygen mask, CPAP, and intubation. A more detailed analysis of the different treatment strategies in COVID-19 patients is shown in Appendix A. Patients receiving oxygen therapy showed a non-significant trend (*p* = 0.055) of reduced serum HDL-C at baseline, while the cholesterol efflux capacity was significantly lower (*p* = 0.012) (Figure 3A,C). Of particular interest, the activity of LCAT was also reduced in the oxygen-treated patients (*p* = 0.005), suggesting a disturbance in HDL maturation (Figure 3B). Further, the AE-activity of HDL-associated PON1 was decreased in the patients receiving ventilatory assistance at baseline (*p* = 0.041) (Figure 3D), similar to the reduction in the anti-oxidative capacity (*p* = 0.014) (Figure 3E). While serum levels of SAA were higher in the oxygen-treated patients (*p* = 0.034) (Figure 3G), the anti-inflammatory activity of apoB-depleted serum was not different between the groups (*p* = 0.219) (Figure 3F). The results suggest that patients needing oxygen therapy show profound alterations in HDL quantity, composition, metabolism, and in functional parameters.

### 3.4. HDL Cholesterol Efflux Capacity Is Inversely Associated with Mortality Risk in COVID-19 Patients

Death occurred in 31% of the COVID-19 patients. Using Kaplan–Meier and Cox regression analyses, we next assessed whether HDL-related functional parameters were associated with an increased risk of death.

For Kaplan–Meier analysis, patients were stratified into two groups for each variable: (1) low and (2) high levels/activity of the parameter based on the median (Figure 4). The log-rank test was calculated to analyze differences between the two groups. Kaplan–Meier analyses revealed that patients having low cholesterol efflux capacity had an increased risk of mortality (*p* = 0.033) (Figure 4C). No significant association with risk of death was observed for groups with low levels of AE-activity, AO-capacity, anti-inflammatory capacity, and SAA with the outcome (Figure 4D–G). Interestingly, we observed that low serum HDL-C concentration was not significantly associated with an increased risk of the outcome (*p* = 0.316) (Figure 4A), however we found a trend for an association of low LCAT activity with increased mortality risk (*p* = 0.060) (Figure 4B).

To evaluate whether the association of HDL-functionalities at baseline with mortality in COVID-19 patients is independent of serum HDL-C levels, we used a continuous, multivariable Cox regression model adjusted for age, gender, and HDL-C (Figure 5).

Of particular interest, we observed that cholesterol efflux capacity was inversely associated with risk of mortality in the patients (HR per 1-SD increase in cholesterol efflux capacity was 0.38, with 95% CI of 0.15 to 0.96), while AE-activity, AO-capacity, and the anti-inflammatory activity were not associated.

## 4. Discussion

In the present study, we examined changes in qualitative HDL parameters in COVID-19 patients compared with patients with non-COVID pneumonia. We found that patients with COVID-19 had significantly lower serum activities of LCAT and markedly impaired HDL functionality, such as reduced cholesterol efflux capacity, AE-activity of PON1, and AO-capacity, when compared to non-COVID pneumonia patients. Activity of LCAT on nascent or lipid-poor HDL particles composed of phospholipid, cholesterol, and apoA-I catalyzes the esterification of cholesterol in HDL [32,33]. A similar decrease in LCAT activity has been observed in other infectious diseases and may be due to infection-related impairment of liver function, as LCAT is mainly produced by the liver [21,35]. LCAT is a key enzyme in lipoprotein metabolism that enables the maturation of HDL particles, and this activity may help to drive cholesterol efflux. A very important finding of the present study was that the cholesterol efflux capacity of HDL was significantly decreased in COVID-19 patients. Moreover, HDL cholesterol efflux capacity was the strongest predictor of COVID-19 mortality in the survival analysis, independent of HDL-C levels. The cholesterol efflux capacity of HDL is directly linked to the ability of HDL to modulate the cholesterol content of the lipid rafts of the plasma membrane [14,16]. Most importantly, lipid rafts on host cell plasma membranes have an important role in viral entry and budding [36]. Cholesterol-depleting molecules, such as methyl-β-cyclodextrin, inhibit the cellular entry of coronaviruses [36,37]. A recent study demonstrated that plasma membrane cholesterol increased the trafficking of angiotensin-converting enzyme 2 (ACE2) to lipid rafts, thereby increasing SARS-CoV-2 infection [17]. On the other hand, the removal of plasma membrane cholesterol reduced the levels of ACE2 and the furin protease in lipid rafts, thereby reducing SARS-CoV-2 infection [17]. Thus, plasma membrane cholesterol is required for proper trafficking and localization of receptors that facilitate SARS-CoV-2 infection. In addition, the efflux of membrane cholesterol and ingested cholesterol and possibly oxidized phospholipids and sterols from phagocytes likely helps maintain the viability of immune cells, phagocytes and endothelial cells [38]. This clearly suggests that a high cholesterol efflux capacity of HDL could interfere with SARS-CoV-2 infection by the efficient removal of lipid raft cholesterol. Administration of recombinant HDL in a severe COVID-19 patient has already been suggested as a potential strategy to be investigated in COVID-19 [39]. It may therefore be highly interesting to test cholesterol efflux modulating therapies in the prevention and treatment of COVID-19.

Interestingly, CRP levels showed a strong negative association with cholesterol efflux capacity (r_S_ = −0.585, *p* < 0.0001), illustrating that inflammation strongly impairs the protective properties of HDL. CRP levels also negatively associated with HDL-C, LCAT activity, and AE-activity of PON1, and strongly with SAA levels. Acute phase proteins such as SAA have been shown to displace apoA-I from HDL, leading to a faster catabolism of apoA-I in the liver and kidney [19,40]. In our study, we observed no difference in SAA levels between COVID-19 patients and subjects without COVID pneumonia, suggesting that both groups had comparable inflammatory status. In a recent study, SAA levels were found to be increased by more than 50% in hospitalized COVID-19 patients compared with patients with only mild symptoms, emphasizing the association with disease severity [26]. As such, the HDL proteome in COVID-19 patients also suggests profound inflammatory remodeling of HDL associated with the severity of COVID-19 infection [26].

Interestingly, serum creatinine levels correlated significantly with the AO-capacity in our cohort. A similar association was also observed in patients with chronic kidney disease [41]. Given that the antioxidant capacity of serum is also dependent on low molecular weight hydrophilic antioxidants, decreased clearance of these hydrophilic antioxidants may explain the inverse association of AO capacity serum with renal function [41].

PON1 is an important contributor to HDL’s anti-oxidative and anti-inflammatory activities. In particular, PON1 hydrolyzes oxidized lipids and is able to neutralize homoserine lactones from pathogenic bacteria [42,43,44]. Interestingly, in sepsis patients, the AE-activity of PON1 was suggested as an independent predictor of 28-day and ICU mortality in multivariable analyses [35]. It has already been shown that in COVID-19 patients, apoA-I and PON1 are less abundant, whereas the content of SAA and alpha-1 antitrypsin was higher [27]. This is in line with the fact that during infections, the half-life of apoA-I, the major apolipoprotein of HDL, is markedly decreased, leading to reduced serum apoA-I and HDL-C levels [19,45,46].

Apart from the well-known function of HDL to promote reverse cholesterol transport, the particles are also able to bind and neutralize LPS from Gram-negative bacteria, thereby preventing systemic endotoxemia [8,47]. Moreover, HDL also displays pleiotropic effects on the endothelium by activating endothelial nitric oxide synthase, which leads to reduced vascular tension [48]. The anti-inflammatory properties of HDL comprise their effect on inhibiting the expression of adhesion molecules, thus preventing monocyte transmigration through the endothelium [49,50]. HDL further inhibits the production of pro-inflammatory cytokines by modulating nuclear factor κB (NF-κB) and the peroxisome proliferator-activated receptor γ [51]. In the present study, we found no difference in the ability of HDL to suppress lipopolysaccharide induced NF-κB expression in monocytes between COVID-19 and non-COVID pneumonia patients. In a study comparing the anti-inflammatory function of HDL in COVID-19 patients with healthy controls, the authors demonstrated less protection for endothelial cells stimulated with TNFα [27]. However, in contrast to our study, this study compared HDL functionalities of COVID-19 patients with a healthy cohort. Another interesting observation of our study was that patients who required oxygen therapy had even more profound changes in HDL functional parameters.

The strengths of this study are that we assessed multiple functional parameters of HDL in this study cohort, including AO-capacity and anti-inflammatory activities. To the best of our knowledge, the present study is a first in assessing HDL cholesterol efflux capacity in COVID-19 patients. Another strength is the control cohort—patients who had non-COVID pneumonia and presented similar levels of inflammatory markers. Further, we performed survival analyses, as information on time and outcome was available.

The present study has some limitations mainly related to the small sample size of the analyzed cohort, even though it was sufficiently powered to support the overall changes in serum HDL functionalities between subjects with or without COVID-19. An additional limitation may be the observational nature of the study. In this regard, it will be interesting in the future to add mechanistic insights to our findings by evaluating detailed changes in HDL structure and composition that would explain the alterations in HDL functions.

## 5. Conclusions

To conclude, COVID-19 patients showed profound differences in the functionality of HDL besides the quantity of HDL-C. We observed an inverse association of cholesterol efflux capacity with mortality risk, independent of serum HDL-C levels. Our findings could help to develop new pharmacological therapies and may help clinicians to predict the severity of COVID-19 infection. Further studies are needed to determine the potential role of lipid-modulating therapies in the prevention and management of COVID-19.

## Figures and Tables

**Figure 1 antioxidants-11-01858-f001:**
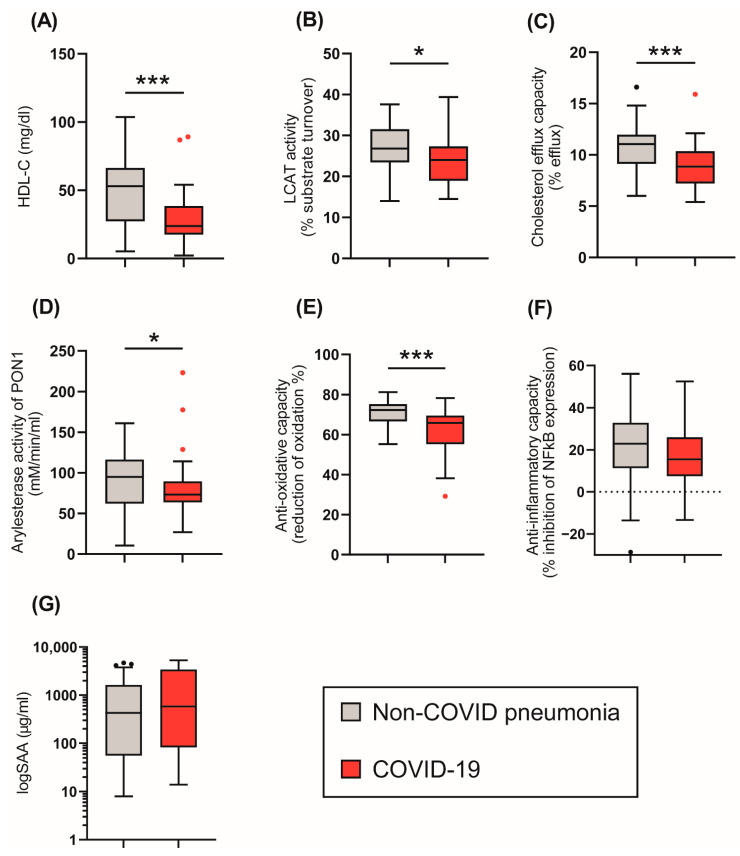
Tukey boxplots of HDL-related parameters in the COVID-19 (*n* = 48) and control cohort (*n* = 32) at baseline. (**A**) HDL-cholesterol levels (mg/L); (**B**) LCAT activity (% substrate turnover); (**C**) Cholesterol efflux capacity (%); (**D**) arylesterase activity of PON1 (mM/min/mL serum); (**E**) anti-oxidative capacity (% reduction of oxidation); (**F**) anti-inflammatory capacity (% inhibition of NFκB expression); (**G**) logSAA (µg/mL). Differences between the two groups were analyzed by Mann–Whitney U test. Outliers are represented by black and red dots. * *p* < 0. 05, *** *p* < 0.001.

**Figure 2 antioxidants-11-01858-f002:**
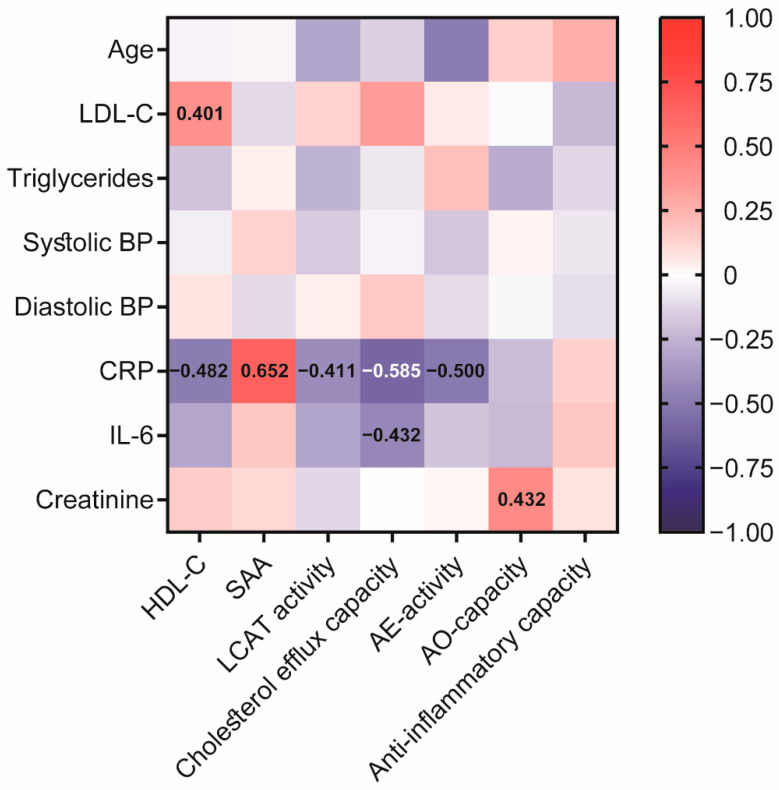
Heatmap representing correlations between HDL-related parameters and clinical data in the COVID-19 patients at baseline. Each cell of the heatmap represents pairwise Spearman correlation between the two parameters indicated in the respective row and column. Correlations that reached significance after Bonferroni correction are indicated with the corresponding Spearman correlation coefficient. Non-significant correlations can still be inferred from the color but are not explicitly indicated. BP, blood pressure; CRP, C-reactive protein; IL-6, interleukin-6; LCAT, lecithin cholesteryl acyltransferase; SAA, serum amyloid A.

**Figure 3 antioxidants-11-01858-f003:**
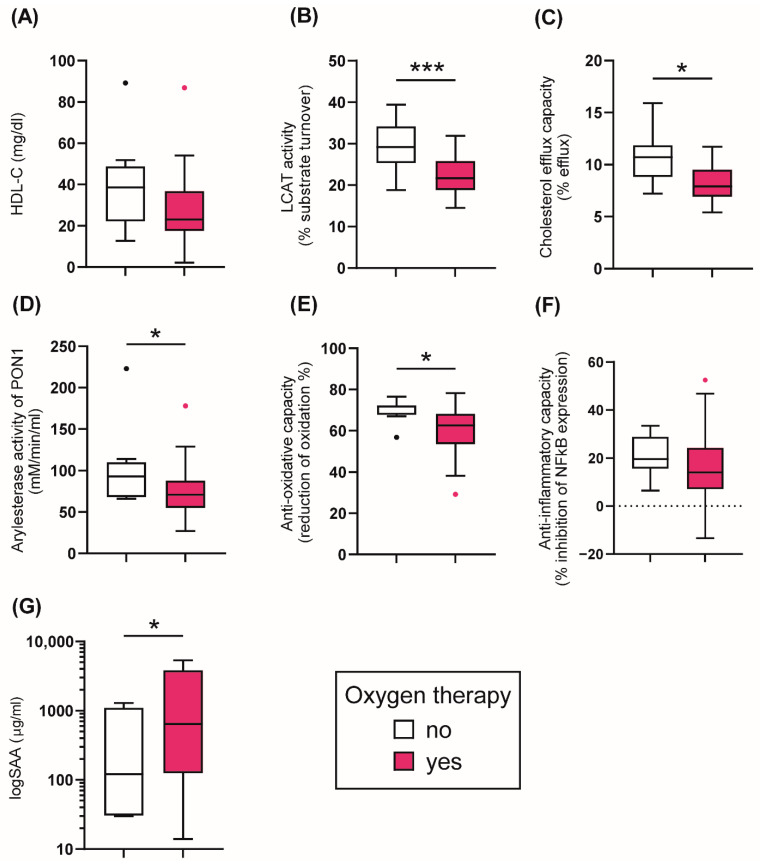
Tukey boxplots of HDL-related parameters in COVID-19 patients (*n* = 48) stratified by different treatment options. Oxygen therapy included treatment with oxygen mask, oxygen with CPAP and mechanical ventilation. (**A**) HDL-cholesterol levels (mg/l); (**B**) LCAT activity (% substrate turnover); (**C**) Cholesterol efflux capacity (%); (**D**) arylesterase activity of PON1 (mM/min/mL serum); (**E**) anti-oxidative capacity (% reduction of oxidation); (**F**) anti-inflammatory capacity (% inhibition of NFκB expression); (**G**) logSAA (µg/mL). Differences between the two groups were analyzed by the Mann–Whitney U test. Outliers are represented by black and red dots. * *p* < 0.05, *** *p* < 0.001.

**Figure 4 antioxidants-11-01858-f004:**
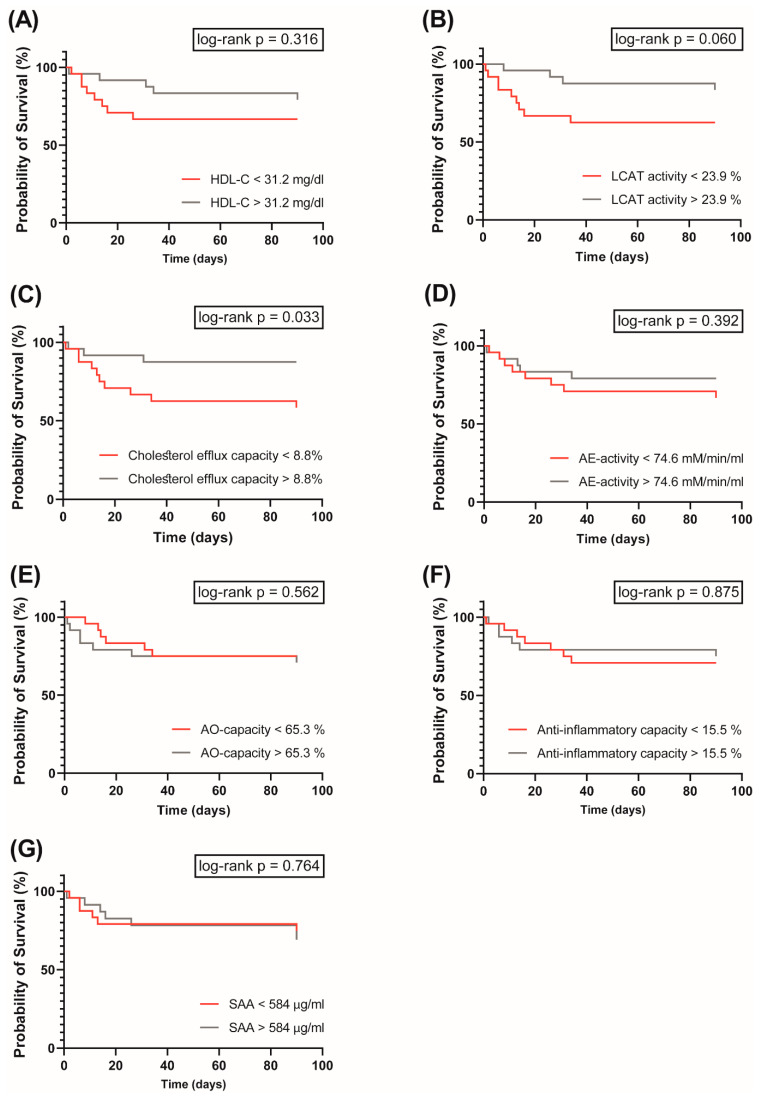
Kaplan–Meier analyses with subsequent log-rank test (endpoint death) in the COVID-19 cohort (*n* = 48) for HDL-C (**A**), LCAT activity (**B**) cholesterol efflux capacity (**C**), AE-activity of PON1 (**D**), AO-capacity (**E**), anti-inflammatory capacity (**F**) and SAA (**G**) are shown. Patients were classified as high (grey line) or low (red line) depending on whether their HDL-related parameters were above or below the median.

**Figure 5 antioxidants-11-01858-f005:**
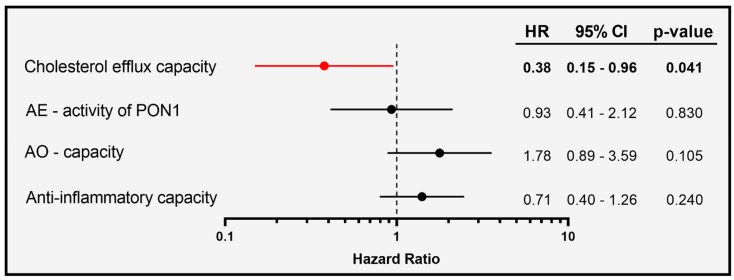
Survival hazard ratios (HRs) per 1 SD increase and 95% confidence intervals (CI) derived from Cox regression analyses. Model was adjusted for age, gender, and HDL-C levels.

**Table 1 antioxidants-11-01858-t001:** Baseline characteristics of the study cohort.

	COVID-19 Patients (*n* = 48)	Non-COVID Pneumonia Patients (*n* = 32)	*p*-Value
Age (years)	68 (56–80)	74 (53–81)	0.529
Female sex	25 (52%)	18 (56%)	0.714
Total cholesterol (mg/dL)	158 (126–197)	205 (162–244)	**0.004**
LDL-cholesterol	89 (65–117)	129 (93–160)	**0.005**
HDL-cholesterol (mg/dL)	23.9 (17.6–38.6)	53.0 (27.4–66.4)	**<0.001**
Triglycerides (mg/dL)	147 (101–228)	138 (79–184)	0.137
CRP (mg/L)	35.4 (12.3–77.0)	52.4 (4.6–91.9)	0.932
IL-6 (ng/L)	38.0 (15.7–116.0)	42.1 (7.1–97.0)	0.415

Baseline characteristics of the study cohort. Data are presented as median (Q1–Q3). Differences between COVID-19 and non-COVID-19 pneumonia patients were assessed by Mann–Whitney U test. *n*, number of subjects; LDL, low-density lipoprotein; HDL, high-density lipoprotein; CRP, C-reactive protein; IL-6, interleukin-6.

## Data Availability

Data are contained within the article.

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
