# Peer review of "Low HDL Cholesterol Efflux Capacity Indicates a Fatal Course of COVID-19"

_antioxidants, 2022, doi:10.3390/antiox11101858_

Round 1
Reviewer 1 Report
This is an interesting report showing a significant association between reduced cholesterol efflux capacity and COVID-19 mortality. Of particular interest, the authors used a control cohort of non-COVID-19 pneumonia patients, thus showing that a severe course of COVID-19 requiring hospitalization and oxygen therapy uniquely reduces HDL cholesterol efflux capacity compared to other types of pneumonia. While the patient cohort is small, the data are of significant interest.
There are some minor improvements that would improve clarity and readability of the report.
Figures:
Where p-values are shown, please use the asterisk system (e.g., * = P<0.05, **=P<0.01, etc.). Not all of the p-values indicated on the figures rise to the level of significance, and so it would aid the reader to see the values that are truly significant, and be able to judge between levels of significance.
Black and red dots on the figures (applies to Figs. 1 and 3) - I am assuming these indicate outliers, but it would help if the authors stated this explicitly.
Supplemental Figures: There is no reference to these figures in the body of the text. If they contribute to the report, then there should be mention of how they contribute. If they do not, they shouldn't be included. As with Figs. 1 & 3, significance should be indicated using the asterisk system.
Text:
Lines 185-188: This appears to be associated with a figure or a table (Table 1?) but needs some cleaning up and clarity.
Line 357: Delete the comma.
Author Response
Response:
This is an interesting report showing a significant association between reduced cholesterol efflux capacity and COVID-19 mortality. Of particular interest, the authors used a control cohort of non-COVID-19 pneumonia patients, thus showing that a severe course of COVID-19 requiring hospitalization and oxygen therapy uniquely reduces HDL cholesterol efflux capacity compared to other types of pneumonia. While the patient cohort is small, the data are of significant interest.
There are some minor improvements that would improve clarity and readability of the report.
We are pleased that our manuscript was favorably received by the reviewer and are happy to consider the helpful comments.
Figures:
Where p-values are shown, please use the asterisk system (e.g., * = P<0.05, **=P<0.01, etc.). Not all of the p-values indicated on the figures rise to the level of significance, and so it would aid the reader to see the values that are truly significant, and be able to judge between levels of significance.
According to the reviewer’s suggestion, we have changed the p-values in figure 1 and 3 to asterisks where necessary.
Black and red dots on the figures (applies to Figs. 1 and 3) - I am assuming these indicate outliers, but it would help if the authors stated this explicitly.
According to the reviewer’s suggestion, we have added the statement “Outliers are represented by black and red dots” in the figure legends.
Supplemental Figures: There is no reference to these figures in the body of the text. If they contribute to the report, then there should be mention of how they contribute. If they do not, they shouldn't be included. As with Figs. 1 & 3, significance should be indicated using the asterisk system.
We thank the reviewer you for this comment. We have decided to remove Supplementary. Figure 2 as it is not necessary for our study and have added a reference to Suppl. Figure 1 (line 236).
Text:
Lines 185-188: This appears to be associated with a figure or a table (Table 1?) but needs some cleaning up and clarity.
Thank you for pointing out this problem. We have now added a title to the footer of the table and reformatted it.
Line 357: Delete the comma.
The typo has been corrected.
Reviewer 2 Report
In this interesting paper, the authors investigated the HDL cholesterol efflux capacity, HDL anti-inflammatory and antioxidant properties, and changes in HDL 21 composition and metabolism in COVID-19 (n=48) and non-COVID pneumonia patients (n=32). Cholesterol efflux capacity (CEC) is an in-vitro assay that measures the ability of an individual's HDL to promote cholesterol efflux from cholesterol donor cells such as macrophages. CEC of HDL is a predictor of cardiovascular risk independent of HDL-C levels and is supposed to be an additional biomarker of COVID-19 morbidity and mortality.
The authors found that COVID-19 infection markedly reduced the activity of lecithin-cholesteryl-acyltransferase and functional parameters of HDL, such as the cholesterol efflux capacity, arylesterase activity of 24 paraoxonase 1 and anti-oxidative capacity of apoB-depleted serum when compared to non-COVID pneumonia at baseline, paralleled by markedly reduced levels of HDL-cholesterol. Of particular interest, low HDL cholesterol efflux capacity was associated with increased mortality risk in COVID-19 patients, independent of HDL-C levels.
The authors discussed their findings stating that their results highlight the profound effects of COVID-19 infection on HDL function, metabolism and composition. In particular, authors speculated that low HDL cholesterol efflux capacity indicates a fatal course of COVID-19, independent of HDL-cholesterol levels.
The paper is potentially suitable for publication in Antioxidants, however, to reach the level of the scientific quality of the journal it should be improved in the methodology and individual recruitment. In particular:
- the n of the enrolled individuals is low since includes men and women
- where are the healthy controls? I do not believe that non-COVID pneumonia patients are the appropriate controls for COVID-19 patients since the recruited individuals are not young and during aging, the serum lipids’ profile may be disrupted.
- the gender effect should be analyzed in some way. I do suggest a two-way ANOVA (disease x gender).
- data should be carefully correlated per the age of the enrolled subjects since the groups’ age range is quite large (53-81 in the pneumonia group, 56-80 in the COVID group).
Author Response
In this interesting paper, the authors investigated the HDL cholesterol efflux capacity, HDL anti-inflammatory and antioxidant properties, and changes in HDL 21 composition and metabolism in COVID-19 (n=48) and non-COVID pneumonia patients (n=32). Cholesterol efflux capacity (CEC) is an in-vitro assay that measures the ability of an individual's HDL to promote cholesterol efflux from cholesterol donor cells such as macrophages. CEC of HDL is a predictor of cardiovascular risk independent of HDL-C levels and is supposed to be an additional biomarker of COVID-19 morbidity and mortality.
The authors found that COVID-19 infection markedly reduced the activity of lecithin-cholesteryl-acyltransferase and functional parameters of HDL, such as the cholesterol efflux capacity, arylesterase activity of 24 paraoxonase 1 and anti-oxidative capacity of apoB-depleted serum when compared to non-COVID pneumonia at baseline, paralleled by markedly reduced levels of HDL-cholesterol. Of particular interest, low HDL cholesterol efflux capacity was associated with increased mortality risk in COVID-19 patients, independent of HDL-C levels.
The authors discussed their findings stating that their results highlight the profound effects of COVID-19 infection on HDL function, metabolism and composition. In particular, authors speculated that low HDL cholesterol efflux capacity indicates a fatal course of COVID-19, independent of HDL-cholesterol levels.
The paper is potentially suitable for publication in Antioxidants, however, to reach the level of the scientific quality of the journal it should be improved in the methodology and individual recruitment. In particular:
We thank the reviewer for reviewing our manuscript and providing the feedback and suggestions for improvement.
- the n of the enrolled individuals is low since includes men and women
We agree with the reviewer that further larger studies are needed to confirm our observations, however, the n-number of our study cohort was sufficiently powered to support the overall changes in serum HDL functionalities between COVID-19 patients and non-COVID pneumonia patients. In addition, the two groups were matched in age and gender.
- where are the healthy controls? I do not believe that non-COVID pneumonia patients are the appropriate controls for COVID-19 patients since the recruited individuals are not young and during aging, the serum lipids’ profile may be disrupted.
In our opinion, healthy controls would not be an adequate control cohort for the following reasons. We and many other groups have clearly shown that inflammation leads to strong changes in HDL functionality (DOI: 10.1016/j.pharmthera.2012.12.001). Because we wanted to detect COVID-19-related changes, NON-COVID pneumonia patients were our control cohort because their disease has a strong inflammatory component triggered by in the same target organ of an infectious disease. Furthermore, both groups suffer from oxygen deficits due to hampered oxygen uptake in the lungs, and show a high oxidative stress level. Hence, the inflammatory/oxidative bias to HDL composition and function is comparable. If we had taken healthy subjects as controls, we would not know whether specifically COVID-19 induced effects triggered the changes in HDL functionalities.
This was also acknowledged very positively by reviewer 1, who noted in his comments, “of particular interest, the authors used a control cohort of non-COVID-19 pneumonia patients, thus showing that a severe course of COVID-19 requiring hospitalization and oxygen therapy uniquely reduces HDL cholesterol efflux capacity compared to other types of pneumonia”.
- the gender effect should be analyzed in some way. I do suggest a two-way ANOVA (disease x gender).
We kindly ask the reviewer to note that the groups were gender-matched and therefore gender should not affect group differences. Moreover, our calculated survival hazard ratios derived from Cox regression analyses (Figure 5) were adjusted for age, gender and HDL-C levels.
According to the reviewer’s suggestion, we performed two-way ANOVA analysis to identify significant effects of the interaction of disease and gender on parameters of HDL function. However, only for LCAT activity, a significant effect was found (p=0.037). As shown in the table, for all the other assessed HDL-related parameters, no significant effect of the interaction between gender x disease was observed.
We believe that the effects of gender on parameters of HDL function in COVID-19 patients have be investigated in a much larger study cohort to have enough statistical power to draw firm conclusions.
|
|
|
df |
Mean Square |
p-value |
|
Cholesterol efflux capacity |
Gender |
3 |
0.309 |
0.807 |
|
Disease |
1 |
85.228 |
<0.001 |
|
|
Gender x disease |
1 |
10.272 |
0.162 |
|
|
HDL-cholesterol |
Gender |
3 |
903.697 |
0.184 |
|
Disease |
1 |
9185.756 |
<0.001 |
|
|
Gender x disease |
1 |
298.135 |
0.444 |
|
|
Anti-oxidative capacity |
Gender |
3 |
32.760 |
0.529 |
|
Disease |
1 |
1356.797 |
<0.001 |
|
|
Gender x disease |
1 |
256.214 |
0.081 |
|
|
AE-activity of PON1 |
Gender |
3 |
0.079 |
0.994 |
|
Disease |
1 |
3371.353 |
0.104 |
|
|
Gender x disease |
1 |
586.214 |
0.495 |
|
|
LCAT activity |
Gender |
3 |
0.862 |
0.875 |
|
Disease |
1 |
221.764 |
0.013 |
|
|
Gender x disease |
1 |
154.000 |
0.037 |
|
|
Anti-inflammatory activity |
Gender |
3 |
932.220 |
0.064 |
|
Disease |
1 |
40.296 |
0.697 |
|
|
Gender x disease |
1 |
196.837 |
0.391 |
|
|
SAA |
Gender |
3 |
1110503.464 |
0.549 |
|
Disease |
1 |
6596593.225 |
0.147 |
|
|
Gender x disease |
1 |
5781180.258 |
0.174 |
- data should be carefully correlated per the age of the enrolled subjects since the groups’ age range is quite large (53-81 in the pneumonia group, 56-80 in the COVID group).
We again kindly ask the reviewer to note that the groups were age-matched and therefore age should not affect group differences. Moreover, our calculated survival hazard ratios derived from Cox regression analyses were adjusted for age, gender and HDL-C levels.
As shown in Figure 2 of our manuscript, age of the COVID-19 patients was correlated with HDL-related parameters. After Bonferroni correction, only AE-activity of PON1 was significantly inverse associated with age (Figure 2). For all the other HDL-related parameters, no correlation with age was found.
Round 2
Reviewer 2 Report
I found the paper improved, and the answers of the authors to my comments were convincing. The work is fine right now.